# Capillary Blood Gas in Children Hospitalized Due to Influenza Predicts the Risk of Lower Respiratory Tract Infection

**DOI:** 10.3390/diagnostics12102412

**Published:** 2022-10-05

**Authors:** August Wrotek, Teresa Jackowska

**Affiliations:** 1Department of Pediatrics, Centre of Postgraduate Medical Education, Marymoncka 99/103, 01-813 Warsaw, Poland; 2Department of Pediatrics, Bielanski Hospital, Cegłowska 80, 01-809 Warsaw, Poland

**Keywords:** influenza, children, pneumonia, capillary blood gas, acidosis, hypercapnia

## Abstract

Background: Influenza may impair respiratory exchange in the case of lower respiratory tract infections (LRTIs). Capillary blood gas (CBG) reflects arterial blood values but is a less invasive method than arterial blood sampling. We aimed to retrospectively verify the usefulness of CBG in pediatric influenza. Material and methods: CBG parameters (pH, pCO_2_, pO_2_, SatO_2_) in laboratory confirmed influenza cases hospitalized in 2013–2020 were verified in terms of LRTI, chest X-ray (CXR) performance, radiologically confirmed pneumonia (CXR + Pneumonia), prolonged hospitalization, and intensive care transfer. A theoretical CBG-based model for CXR performance was created and the odds ratios were compared to the factual CXR performance. Results: Among 409 children (aged 13 days–17 years 3/12, median 31 months), the usefulness of CBG decreased with the age. The SatO_2_ predicted the LRTI with AUC = 0.74 (95%CI: 0.62–0.86), AUC = 0.71 (0.61–0.82), and AUC = 0.602 (0.502–0.702) in children aged <6 months old (mo), 6–23 mo, 24–59 mo, respectively, while pO_2_ revealed AUC = 0.73 (0.6–0.85), AUC = 0.67 (0.56–0.78), and AUC = 0.601 (0.501–0.702), respectively. The pCO_2_ predicted the LRTI most precisely in children <6 months with AUC = 0.75 (0.63–0.87), yet not in older children. A high negative predictive value for CXR + Pneumonia was seen for SatO_2_ < 6 mo (96.7%), SatO_2_ 6–23 mo (89.6%), pO_2_ < 6 mo (94.3%), pO_2_ 6–23 mo (88.9%). The use of a CBG-driven CXR protocol (based on SatO_2_ and pO_2_) would decrease the odds of an unnecessary CXR in children <2 years old (yo) by 84.15% (74.5–90.14%) and 86.15% (66.46–94.28%), respectively. SatO_2_ and pO_2_ also predicted a prolonged hospitalization <6 mo AUC = 0.71 (0.59–0.83) and AUC = 0.73 (0.61–0.84), respectively, and in 6–23 mo AUC = 0.66 (0.54–0.78) and AUC = 0.63 (0.52–0.75), respectively. Conclusions: The CBG is useful mainly in children under two years, predicts the risk of LRTI, and can help exclude the risk of CXR + pneumonia. Children under six months of age represent the group that would benefit the most from CBG. A CBG-based protocol for the performance of CXR could significantly decrease the number of unnecessary CXRs.

## 1. Introduction

Influenza has a significant impact on the pediatric population due to both the frequency and the severity of the disease. Estimates based on a systematic review of published data report 109.5 million influenza episodes annually only in children under five years of age [1,2,3]. Influenza undergoes seasonal fluctuations with regards to its frequency, although during the SARS-CoV-2 pandemic, the previously known seasonal patterns were altered in various regions, mainly due to nonpharmaceutical interventions [4,5]. The most important preventive measure, influenza vaccination, is globally available. Nonetheless, vaccine hesitancy and the generally low vaccine uptake impede the fight against the spread of influenza, thus facilitating the virus transmission [6,7]. Children play a pivotal role in the transmission of the virus, as their social activity combines household and community contacts, turning children into active influenza vectors [8,9]. Approximately 9% of the cases are complicated with acute lower respiratory tract infections (ALRI) attributable to influenza, which each year translate to over 10.1 million cases, and around 870,000 influenza-related ALRI cases require hospitalizations, thus having a huge impact on the consumption of the health care system resources [1]. Roughly, one in 20 hospital admissions due to ALRI is estimated to be caused by the influenza virus, while prospectively collected data show that 3% of all pneumonia cases in infants is associated with influenza [1,10]. Influenza type A has been related to an increased risk of a life-threatening pneumonia in children (odds ratio = 2.55), but influenza B also carries a significant threat to the population, including the risk of a severe lower respiratory tract infection (LRTI) [11,12]. The huge pathogenic potential of influenza results not only from its virulency, but is enhanced with bacterial coinfections or secondary bacterial infections in close proximity after an influenza episode. Assessments in the pediatric group of patients reveal a bacterial codetection in as many as 80% of cases [13,14,15]. Gas exchange in the course of influenza may be compromised, and potential pathways involved in the regulation of the local immune response and maintenance of gas exchange include the alveolar epithelium (depletion of type I cells might result in serious sequelae), alveolar macrophages, and a number of inter- and intra-cellular crosstalk [16,17,18,19,20]. An altered gas exchange in severe cases may lead to a respiratory failure with the risk of intensive care and a potential fatal outcome [16]. In general estimates, approximately 4% of ALRI deaths in patients younger than five years old might be related to influenza [1]. Influenza may result in ALRI, which possibly affects the gas exchange, and the involvement of the lower respiratory tract is often suspected in hospitalized patients. Thus, a prompt diagnosis of lower respiratory tract involvement is crucial for adequate management, whereas an assessment of the patient’s gas exchange parameters might be helpful in the prediction of ALRI [21].

Arterial blood gas (ABG) sampling is a golden standard in the assessment of the patient’s gas exchange status, but this procedure carries the risk of complications, and as such, is performed almost exclusively in intensive care unit (ICU) settings, not in general pediatric wards [22]. Nonetheless, there is also a promising alternative for ABG, namely, the capillary blood gas (CBG) that measures the gas balance parameters in the peripheral blood taken from the fingertip, earlobe or heel. The CBG allows to indirectly assess the arterial blood parameters without the need for arterial blood sampling at the expense of a slightly lower accuracy, still, this option is much safer and easier to perform [22]. The CBG is also more reliable than venous blood gas (VBG), and as such might be used even in children with an impaired gas exchange [22].

The published data prove a satisfactory correlation between the CBG and the ABG in terms of pH and partial carbon dioxide pressure (pCO_2_) and shows a slightly lower but still significant correlation regarding the partial oxygen pressure (pO_2_) [22,23,24,25]. In addition, the CBG mirrors the ABG values closely, even in the presence of potential confounding factors. The studies cited above have been performed in the intensive care settings, and irrespectively of the patient’s severe condition, reflected by poor perfusion or hypothermia, showed a strong correlation, with the only exception for hypotension, which may distort the results of the pO_2_ [22,24].

We sought to verify the usefulness of the CBG in children hospitalized due to a laboratory confirmed influenza (LCI) in predicting the risk of a lower respiratory tract involvement, the need for a chest radiograph, the presence of a radiologically confirmed pneumonia, a prolonged hospital stay, and an ICU admission. Moreover, a theoretical CBG-driven model for CXR qualification was created in order to calculate the odds ratios of an unnecessary CXR performance and missed pneumonia cases.

## 2. Materials and Methods

This was an observational retrospective study. Children hospitalized at the Department of Pediatrics at the Bielanski Hospital, Warsaw, between January 2013 and December 2020 were eligible for the study. An electronic database of the medical charts was searched for the following final diagnoses of influenza, according to the International Classification of Diseases, 10th Revision (ICD-10): J10 (or J11) with their extensions; while the J10 codes stand for influenza due to an identified influenza virus, and the J11 stand for an influenza due to an unidentified virus. In order to conduct as comprehensive research as possible, patients with J11 codes were also identified and their charts were revised for the existence of a laboratory confirmation of influenza, however, we included only laboratory confirmed cases. The diagnosis was made upon clinical signs/symptoms of influenza alongside a positive rapid influenza diagnostic test (RIDT) and/or a real-time polymerase chain reaction (RT-PCR), performed in a sample from a nasopharyngeal swab. 

The exclusion criteria consisted of a previously known (or diagnosed during the hospitalization) immune deficiency, diabetes, a history of or an ongoing proliferative disease, hemodynamically significant heart disease, and cystic fibrosis. Patients discharged on parental request were also excluded, as the effect of the treatment was uncertain. 

Only children in whom the CBG was performed were enrolled into the final analysis. 

The CBG was performed in each case immediately after the hospital admission; a sample of capillary blood was drawn after a local disinfection of a finger: a needle was used for an initial puncture and a CBG tube was promptly contacted with the punctured site. No prior warming of the site of incision had been performed. Each sample was verified against the presence of air bubbles and sent to the local laboratory. The analysis was performed with the Roche Cobas b121 and b221 analyzer, Roche Diagnostics Ltd., Switzerland (in the period between 1 January 2013 to 26 January 2016), and with the RAPIDLab 348EX Blood Gas System by Siemens, Siemens Healthcare Diagnostics, Germany (27 January 2016–31 December 2020). The whole analytic process was conducted in accordance with the manufacturers’ instructions. For the purposes of the final analysis, the following CBG parameters were included: pH, partial carbon dioxide pressure (pCO_2_), partial oxygen pressure (pO_2_), and oxygen saturation (SatO_2_).

The anonymized patients’ information included demographical (age, sex), and laboratory data (C-reactive protein (CRP), procalcitonin, white blood cells count (WBC), absolute neutrophil count (ANC)), as well as clinical data (duration of the signs/symptoms prior to the hospitalization, duration of the fever, the highest fever during the disease; pulse oximetry, breath rate and heart rate at admission). 

The patients were assigned to the groups depending on the age: under 6 months of age, 6–23 months of age, 24–59 months of age, and ≥60 months old. The choice of the age groups was based upon the most widely used age-based risk groups of a severe influenza course [3,26]. 

The major end-point consisted of a diagnosis of a lower respiratory tract infection (LRTI) and was based upon the final diagnoses coded according to the ICD-10: J10.0 (influenza due to an identified influenza virus with pneumonia), or a laboratory confirmed influenza J10.x (influenza due to an identified influenza virus with extensions other than pneumonia), together with the pneumonia code (j12.x-j18.x) and/or bronchitis (J20.x) and/or bronchiolitis (J21.x) and/or not-specified LRTI (J22). The secondary endpoints included the performance of a chest X-ray (CXR), the presence of radiologically confirmed pneumonia, the length of the hospital stay, an ICU admission, and a fatal outcome. A theoretical model which assumed a CBG-driven performance of a CXR was created. 

The clinical diagnosis of the LRTI was made upon the clinical signs/symptoms of pneumonia, bronchitis or bronchiolitis. According to the local Polish guidelines on the respiratory tract infection diagnosis and treatment, pneumonia might be diagnosed in the presence of the typical signs and/or symptoms, such as cough, tachypnoea, fever, retraction of the intercostal spaces, a dull percussion note, and abnormalities on the auscultation (crackles or a bronchial murmur), while bronchitis is characterized by cough (productive or unproductive) accompanied by wheezing or rales on auscultation [27]. Bronchiolitis is diagnosed when the first episode of the restriction of bronchioles (expiratory dyspnea, wheezing, rales, and/or hypoxia) is seen in the course of a respiratory tract infection in children under 2 years of age [27].

Radiologically confirmed pneumonia was defined as the presence of clinical signs/symptoms plus a radiological confirmation, irrespective of the character of the CXR findings (interstitial, lobar or mixed); no additional assessment on the correlation with the types of the radiological abnormalities was performed. The chest X-ray performance depended on the individualized decisions made by the team of physicians responsible for the treatment and was not influenced by the study. In the case of a clinical suspicion of pneumonia that was not confirmed with the X-ray, the patient was included into the group of LRTI (if the above conditions were met) but excluded from the analysis of the radiologically confirmed pneumonia. The LOS was considered prolonged if the length of the hospitalization exceeded the median value. The ICU admission was the endpoint itself and no detailed analysis of the type of the respiratory support or length of hospitalization in the ICU was carried out. 

The distribution of continuous data was determined with the Kolmogorov–Smirnov test, according to which a mean with a standard deviation (SD) or a median with an interquartile range (IQR) were used to present the normally or abnormally distributed data, respectively. The corresponding parametric (unpaired Student *t*-test) or non-parametric (Mann–Whitney U test) tests were performed. A ROC (receiver-operating characteristic) curve analysis was performed to calculate the area under the curve (AUC), and the Youden index was used to estimate optimal cut-off values for the CBG parameters in the prediction of the categorical endpoints (LRTI, CXR performance, radiologically confirmed pneumonia, prolonged LOS—for this parameter data was categorized upon the median value). The sensitivity, specificity, positive and negative predictive values (PPV and NPV) were computed for the prediction of a LRTI and for the radiologically confirmed pneumonia. Based upon the negative predictive values for radiologically confirmed pneumonia, the best performing CBG parameters were chosen for the theoretical model of CBG-driven CXR protocol. In such a protocol, the optimal cut-off values were applied to decide on the performance of the CXR. The odds ratios (in comparison to the lack of a CBG-driven protocol) of the unnecessarily performed CXR (i.e., a negative result) and the/omitted radiologically confirmed pneumonia cases with 95% confidence intervals (95%CI) were calculated. The decrease in the odds of an unnecessary CXR (e.g., the one that did not reveal pneumonia) was calculated by subtracting 1 from the odds ratio and its 95% CI. Additionally, a Spearman’s rank test calculated the correlation between the CBG parameters. The *p* value lower than 0.05 was assumed to be statistically significant. The statistical analysis was performed with Statistica 13.1 software (Statsoft, Tulsa, OK, USA). The study obtained the approval by the Ethics Committee at the Centre of Postgraduate Medical Education in Warsaw, Poland (approval number 141/PB/2020 issued on 9 December 2020). Due to the retrospective character of the analysis, the patient’s and parents’ or tutors’ consent was waived.

## 3. Results

A total of 495 children were hospitalized due to influenza in the period between 2013 and 2020. One child was discharged upon parental request, and the CBG was performed in 409 cases (82.6%) which finally created the study group (213 boys and 196 girls) (Figure 1). The patients’ age varied from 13 days to 17 years and three months, with a median of 31 months; the patients’ distribution into the age groups was as follows: <6 mo—71 patients (17%), 6–23 mo—97 patients (24%), 24–59 mo—132 patients (32%), ≥60 mo—109 patients (27%). 

Children younger than two years old with LRTIs presented with a lower pH (7.42 vs. 7.447, *p* = 0.025 in <6 mo group and 7.426 vs. 7.445, *p* = 0.046 in 6–23 mo), higher pCO_2_ (37.57 vs. 32.07, *p* < 0.01 in <6 mo), lower pO_2_ (median 52.4 vs. 60.9, *p* = 0.002 in <6 mo and median 65.8 vs. 71.2, *p* < 0.01 in 6–23 mo), and lower SatO_2_ (88.2 vs. 91.7, *p* < 0.01 in <6 mo and median 93.45 vs. 95.1, *p* < 0.01 in 6–23 mo) (Table 1). Of note, children above two years showed no statistically significant differences in the CBG parameters, regardless of the lower respiratory tract involvement. 

The ROC curve analysis revealed that the number of the CBG parameters able to predict the LRTI decreased with the age. While in <6 mo each of the analyzed CBG parameters showed statistically significant AUC, in the 6–23 mo group, the pH, pO_2_, and SatO_2_ correlated with the LRTI, and in 24–59 mo, only the pO_2_ and SatO_2_ were associated with the LRTI, and none of the parameters were significant in patients older than five years (Figure 2). 

The pCO_2_ had the highest area under the curve for the prediction of the LRTI in children <6 mo (AUC = 0.75, 95%CI: 0.63–0.87, *p* < 0.01), but remained insignificant in older age groups (Table 2). The pH, on the other hand, was relevant both in children aged <6 mo and 6–23 mo, but had a much lower AUC (0.65, 95%CI: 0.52–0.78, *p* = 0.02, and 0.62, 95%CI: 0.51–0.73, *p* = 0.33, respectively). The SatO_2_ showed high a AUC in children <6 mo, 6–23 mo, and lower in 24–59 mo (AUC = 0.74, 95%CI: 0.62–0.86, *p* < 0.01, AUC = 0.71, 95%CI: 0.61–0.82, *p* < 0.01, and AUC = 0.602, 95%CI: 0.502–0.702, *p* = 0.045, respectively), while a slightly lower AUC was observed in the case of pO_2_ in the same age groups (AUC = 0.73, 95%CI; 0.6–0.85, *p* < 0.01, AUC = 0.67, 95%CI: 0.56–0.78, *p* < 0.01, and AUC = 0.601, 95%CI; 0.501–0.702, *p* = 0.0475, respectively). 

Optimal cut-off values for children <6 mo were calculated and equaled: pH = 7.442 (sensitivity = 73.9%, specificity = 54.2%, PPV = 43.6%, and NPV = 81.3%), pCO_2_ = 36.1 mmHg (sensitivity = 69.6%, specificity = 72.9%, PPV = 55.2%, and NPV = 83.3%), pO_2_ = 58 mmHg (sensitivity = 73.9%, specificity = 60.4%, PPV = 47.2%, and NPV = 82.9%), SatO_2_ = 93% (sensitivity = 91.3%, specificity = 50.%, PPV = 46.7%, and NPV = 92.3%) (Table 2). 

In children 6–23 mo, the following cut-offs were established: pH = 7.445 (sensitivity = 82.5, specificity = 47.4%, PPV = 52.4%, and NPV = 79.4%), pO_2_ = 68.6 mmHg (sensitivity = 72.5%, specificity = 59.7%, PPV = 55.8%, and NPV = 75.6%), SatO_2_ = 94% (sensitivity = 72.5%, specificity = 64.9%, PPV = 59.2%, and NPV = 77.1%).

In patients aged 24–59 months, the cut-offs were established at: pO_2_ = 76.6 mmHg (sensitivity = 91.3%, specificity = 29.1%, PPV = 40.8%, and NPV = 86.2%), SatO_2_ = 96.2% (sensitivity = 95.7%, specificity = 23.3%, PPV = 40%, and NPV = 90.9%).

For the prediction of the chest X-ray performance (irrespective of its result), the highest AUC in children < 6 mo was seen for pO_2_ (AUC = 0.7, 95%CI: 0.57–0.83, *p* = 0.022), followed by SatO_2_ (AUC = 0.68, 95%CI: 0.53–0.82, *p* = 0.016), and pCO_2_ (AUC = 0.66, 95%CI: 0.52–0.81, *p* = 0.027) (Table 3). In children 6–23 mo, the highest AUC was observed for SatO_2_ (AUC = 0.75, 95%CI: 0.65–0.85, *p* < 0.01), followed by pO_2_ (AUC = 0.69, 95%CI: 0.59–0.8, *p* < 0.01) and pCO_2_ (AUC = 0.62, 95%CI: 0.51–0.73, *p* = 0.033). No significant associations were found in older children. 

More significant relationships were observed for a radiologically confirmed pneumonia. SatO_2_ demonstrated the highest AUC in both <6 mo and 6–23 mo groups (AUC = 0.76, 95%CI: 0.62–0.89, *p* < 0.01 and AUC = 0.76, 95%CI: 0.66–0.86, *p* < 0.01, respectively), and was followed by pO_2_ (AUC = 0.75, 95%CI: 0.62–0.89, *p* < 0.01 and AUC = 0.7, 95%CI: 0.59–0.8, *p* < 0.01, respectively, Figure 3). An optimal cut-off for SatO_2_ was established: SatO_2_ < 6 mo = 91.6% showed a sensitivity = 93.3%, specificity = 51.8%, PPV = 34.2%, and NPV = 96.7%, while SatO_2_ 6–23 mo = 94% had a sensitivity = 82.1%, specificity = 62.3%, PPV = 46.9%, and NPV = 89.6%; pO_2_ < 6 mo = 58 mmHg showed a sensitivity = 86.7%, specificity = 58.9%, PPV = 36.1%, and NPV = 94.3%, and pO_2_ 6–23 mo = 68.6 mmHg had a sensitivity = 82.1%, specificity = 58%, PPV = 44.2%, and NPV = 88.9%. 

If the calculated cut-off values were to be applied to decide on the CXR performance, the use of SatO_2_ in children under 2 years old (91.6 mmHg <6 mo and 94 mmHg in 6–23 mo) would decrease the odds of an unnecessary CXR by 84.15% (95%CI: 74.5–90.14%, *p* < 0.01), while the pO_2_ use would decrease the odds by 84.58% (95%CI: 75.17–90.42%) (Table 4). The odds of omitting a radiologically confirmed pneumonia were statistically insignificant (with an assumption of no omitted cases when no-CBG protocol was made). Moreover, children with a positive CXR who would not have their CXR performed due to normal SpO_2_ (six children) or pO_2_ (seven children) values, 16.7% and 14.3%, respectively (one child in each group), did not obtain antibiotics, meaning that even a positive CXR did not influence their clinical management. 

The AUC for the extended LOS prediction in <6 mo was 0.73 (95%CI: 0.61–0.84, *p* < 0.01) for pO_2_, followed by SatO_2_ (AUC = 0.71, 95%CI: 0.59–0.83, *p* < 0.01) and pCO_2_ (AUC = 0.64, 95%CI: 0.5–0.77, *p* = 0.045), while in 6–23 mo the highest AUC was observed for SatO_2_ (AUC = 0.66, 95%CI: 0.54–0.78, *p* < 0.01), followed by pO_2_ (AUC = 0.63, 95%CI: 0.52–0.75, *p* = 0.025) (Table 3). Inversely, in children 24–59 mo, the increased pCO_2_ correlated with a shorter LOS (AUC = 0.39, *p* = 0.036). 

Due to the low number of patients transferred to the ICU (one patient), a further analysis on this endpoint would present a low statistical value and was thus not performed. No fatal cases were reported. 

An internal correlation between the CBG parameters was observed between the pH and pCO_2_ (rho = −0.78), pO_2_ (rho = 0.33), and SatO_2_(rho = 0.57) and between the pCO_2_ and pO_2_ (rho = −0.49), SatO_2_ (rho = −0.66).

## 4. Discussion

This study shows that the capillary blood gas has a promising value in the assessment of the risk of a lower respiratory tract involvement in young children hospitalized due to influenza and might possibly prompt a new direction in the assessment of pediatric influenza patients. To the best of our knowledge, this is the first study focusing on the use of the CBG in children with influenza. It needs to be underscored that, except for numerical merit, the use of CBG seems to be promising due to practical implications. While capillary blood sampling is a relatively easy-to-perform and repeatable method, arterial blood sampling is far more invasive, technically difficult, and painful [22]. We did not analyze patients’ compliance nor opinion on arterial versus capillary sampling, but we are convinced that when a less invasive method of comparable value is available, it should be the preferred method. 

Firstly, the influence of the patients’ age needs to be considered; while in children under six months old each of the analyzed CBG parameters exhibited a significant AUC for the prediction of the lower respiratory tract involvement, in older children the predictive value of the pCO_2_, pH, and SatO_2_ or pO_2_ gradually disappeared with the increasing age (at 6–23, 24–59, and over 60 mo, respectively). It is interesting to note that the upper age limit after which the CBG loses its usefulness coincides with the upper age limit of the risk of a severe influenza course [26]. This might be explained by a lower impact of LRTIs on older children; even if the lower respiratory tract is infected, it does not disturb the gas exchange to such a high degree as observed in younger patients.

In general, the SatO_2_ and pO_2_ demonstrate the best performance in the prediction of LRTIs in children under 24 mo. Both SatO_2_ and pO_2_ also showed a statistical significance in children 24–59 mo in the LRTI risk assessment, although its power decreased with age, and became insignificant over the age of five years. In children under 6 mo and 6–23 mo, the AUC was moderate and differed only slightly between SatO_2_ and pO_2_, with a preference for SatO_2_(AUC = 0.74 and 0.73 for <6 mo, or 0.71 and 0.67 for 6–23 mo, respectively), although both parameters might be considered for the purposes of a LRTI prediction. 

It needs to be emphasized that although some doubts on the true reflection of the arterial parameters have been raised in the case of the SatO_2_ and pO_2_, in our group of patients, the capillary SatO_2_ and pO_2_ turned out to be independent predictors of LRTIs, not biased by the uncertainties on its true relationship with the arterial blood parameters. The published data show 65–69% of concordance between capillary and arterial blood in the case of pO_2_ [22,23,24], yet some authors reported ratios as low as r = 0.358 [25] (Harrison et al. 1997). 

The pCO_2_, on the other hand, seems to reflect arterial values more precisely. However, in our series of patients, its use is restricted to the youngest group [22]. The pCO_2_ showed the best performance in infants under 6 mo, but it was insignificant in older children. The previous studies on the pCO_2_ confirmed a strong correlation between the arterial and capillary blood results. Yildildas and colleagues revealed the concordance ratio of 0.988, which is in line with the previous reports showing ratios between r = 0.86, r = 0.9534, and r = 0.955 [22,23,24,25]. Interestingly, although the level of agreement between the CBG and the ABG is also high in the case of pH (a slightly lower correlation—between r = 0.823 and r = 0.903) [22,25], the pH (just as pCO_2_) was relevant to a lower degree than SatO_2_ or pO_2_. There are several possible explanations for these alleged discrepancies. The first one is the group selection, as patients included in our study (except for one) did not present a respiratory failure, and obviously a lower respiratory tract involvement is not unequivocally related to a respiratory failure. Promising results of pCO_2_ and pH use in the prediction of an ICU transfer were observed in children with bronchiolitis [28,29], but not in this group of influenza hospitalizations. In fact, influenza LRTIs may affect oxygen supply to a higher degree than carbon dioxide elimination or acid-base balance. The research by Zhang et al. reported decreased arterial pO_2_ in 73.5% of patients (adults, only one child was included into the study) hospitalized due to H7N9 influenza, while the paCO_2_ was increased only in 3% of cases [30]. Secondly, the age-related differences in the respiratory tract pathophysiology might offer another interpretation: limited compensatory capacities in the youngest group of patients may rapidly lead to an increased pCO_2_, while the older children are not affected so significantly. Bronchiolitis is diagnosed mainly in the first year of life, while influenza patients were seen in each age group. Thirdly, differences between the etiological factors need to be recognized, since the major etiological factor of bronchiolitis is respiratory syncytial virus (RSV), whereas the influenza virus does not result in bronchiolitis so frequently and significant differences in immunological response patterns might be expected [18,31,32]. 

Another interesting aspect is the high negative predictive value in the prediction of a radiologically confirmed pneumonia, which suggests that a CBG-driven protocol might help decrease the number of needlessly performed chest X-rays, the decrease in the study reached almost 85%. Restraining from the CXR on the sole basis of CBG seems to be somehow controversial, but if the CBG was, for example, one of the steps in the verification of the need for a CXR, then children with influenza could be exposed to radiation less frequently. Certainly, this is only preliminary data, and a lack of a radiological confirmation of pneumonia is not equal to a lack of pneumonia, but the practical approach also shows that, in some cases, a positive CXR does not change the clinical management of the patient. 

The relationship between a prolonged hospital stay and the oxygenation parameters in younger children (also an unexpected reverse relationship with pCO_2_ in those aged 24–59 mo) may suggest that oxygenation is more affected in the course of influenza in children, while the carbon dioxide elimination and acid–base balance remain more stable. The association with a respiratory failure and its consecutive stages, however, needs to be established. The internal correlation between the CBG parameters does not let a replacement of one by another, and attention should be paid to their use in the different age groups.

The study may be limited by several factors, including its retrospective character, resulting in the lack of performance of CBG in each patient (although almost 83% of the patients underwent a CBG), and a single-center setting, which impairs a generalization of the results without a further confirmation. Secondly, the optimal study design would consist of a radiological verification of the lower respiratory tract involvement in each patient, yet ethical concerns would make unnecessary exposure to radiation unacceptable. A change of the CBG analyzer needs to be mentioned as well, although an instrument-related bias is unlikely and no significant aberrations are expected. Furtherly, we did not perform influenza subtype or lineage analysis, considering the limited access to such diagnostic tools, even in hospital settings, although clinical differences related to the specific virus should be recognized. Finally, the group preselection might also play a role, and we cannot exclude the usefulness of the CBG in older children, presumably in patients in a more severe condition. However, our results confirm its value only in younger children. 

## 5. Conclusions

In conclusion, we found the CBG a useful tool, mainly in children under two years of age. Children under six months of age represent the group that would benefit the most from CBG. The CBG is able to predict the risk of a lower respiratory tract involvement and, more importantly, to exclude the risk of a radiologically confirmed pneumonia with an approximated 90% NPV. A CBG-based qualification for a CXR performance could significantly decrease the number of unnecessary radiological chest examinations.

## Figures and Tables

**Figure 1 diagnostics-12-02412-f001:**
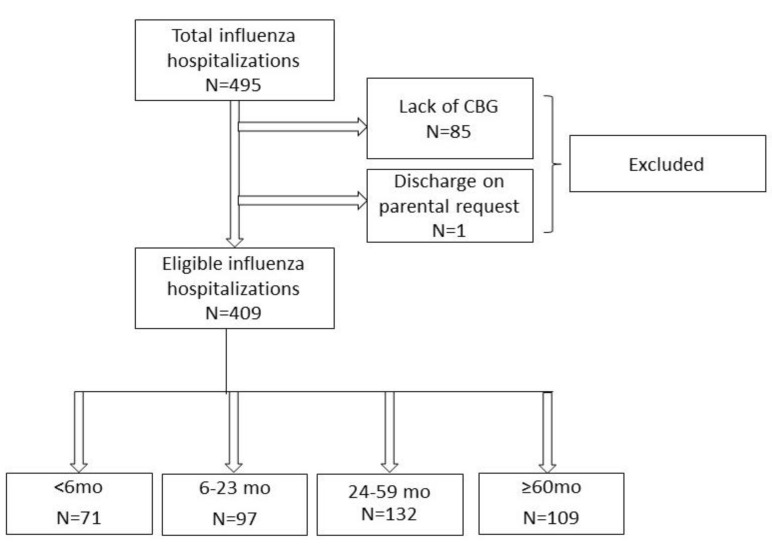
Flow chart of patients in the study.

**Figure 2 diagnostics-12-02412-f002:**
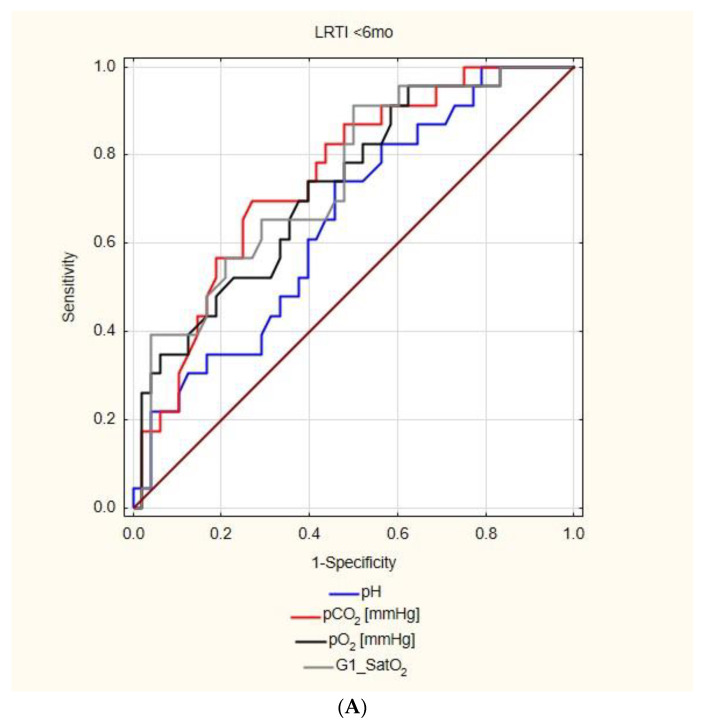
The usefulness of capillary blood gas parameters in the prediction of lower respiratory tract infection (LRTI)- the results of the ROC analysis; (**A**) patients under 6 months of age, (**B**) patients aged 6–23 months old, (**C**) patients aged 24–59 months old.

**Figure 3 diagnostics-12-02412-f003:**
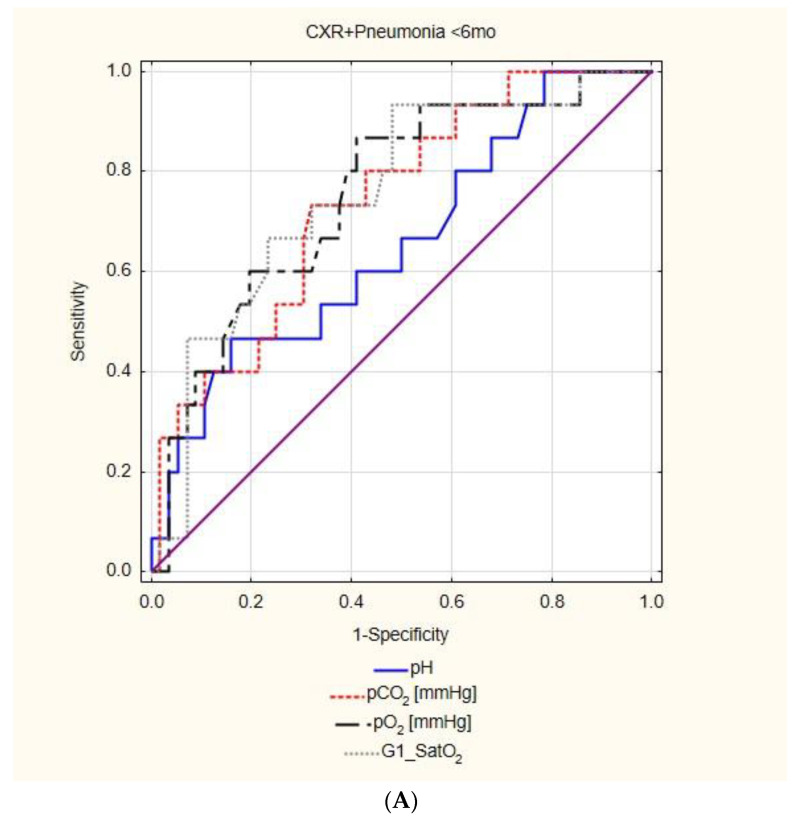
The usefulness of capillary blood gas parameters in the prediction of radiologically-confirmed pneumonia (CXR + Pneumonia)—the results of the ROC analysis; (**A**) patients under 6 months of age, (**B**) patients aged 6–23 months old, (**C**) patients aged 24–59 months old.

**Table 1 diagnostics-12-02412-t001:** Baseline characteristics of the patient study groups according to the lower respiratory tract involvement; the mean and standard deviation (SD) (marked in blue) or median with the interquartile range are shown according to the data distribution; the standard deviation values are followed by the SD. The corresponding test results (*p*-values) are shown: Student *t*-test results are marked with *, while the remaining are the Mann–Whitney U test results. LOS—length of stay, CRP—C-reactive protein, PCT—procalcitonin, WBC—white blood cells, ANC—absolute neutrophil count. Statistically significant results are bolded.

	LRTI			Without LRTI			
<6 mo	Mean/Median with SD or IQR			Mean/Median with SD or IQR			*p*
age [months]	1	0	4	2	1	4	0.703
duration of signs/syndromes [days]	**2**	**1**	**4**	**1**	**1**	**2**	**0.007**
the highest fever [Celsius degrees]	**39.0**	**38.3**	**40.0**	**38.4**	**38.1**	**39.0**	**0.047**
LOS [days]	**10**	**5**	**13**	**5**	**4**	**7**	**0.000**
breath rate [per minute]	**50**	**36**	**60**	**40**	**30**	**40**	**0.005**
heart rate [per minute]	141.24	21.62	(SD)	135.43	17.59	(SD)	0.262 *
CRP [mg/L]	2.20	0.57	11.40	1.25	1.00	4.04	0.393
PCT [ng/mL]	0.14	0.10	0.24	0.15	0.12	0.21	0.582
WBC [*10^3/μL]	**9.90**	**7.40**	**12.70**	**6.90**	**4.66**	**9.70**	**0.015**
ANC [*10^3/μL]	2.25	1.48	3.49	1.99	1.21	2.84	0.092
pH	**7.42**	**0.04**	(SD)	**7.45**	**0.05**	(SD)	**0.025 ***
pCO_2_ [mmHg]	**37.57**	**5.24**	(SD)	**32.07**	**6.56**	(SD)	**0.001 ***
pO_2_ [mmHg]	**52.40**	**47.40**	**60.30**	**60.85**	**52.85**	**67.15**	**0.018**
SatO_2_	**88.21**	**4.04**	(SD)	**91.65**	**4.29**	(SD)	**0.002 ***
6–23 mo	mean/median with SD or IQR			mean/median with SD or IQR			*p*
age [months]	12	9	18	13	9	17	0.657
duration of signs/syndromes [days]	4	1	5	1	1	4	0.053
the highest fever [Celsius degrees]	39.0	38.6	39.5	39.3	38.7	40.0	0.064
LOS [days]	**8**	**6**	**11**	**5**	**4**	**7**	**0.000**
breath rate [per minute]	30	25	40	28	25	30	0.121
heart rate [per minute]	120	115	140	120	110	130	0.345
CRP [mg/L]	6.33	2.72	20.72	4.28	1.20	14.15	0.216
PCT [ng/mL]	**0.35**	**0.18**	**1.03**	**0.20**	**0.13**	**0.54**	**0.026**
WBC [*10^3/μL]	**11.95**	**8.55**	**16.77**	**9.05**	**6.71**	**10.86**	**0.001**
ANC [*10^3/μL]	4.78	2.55	8.92	4.16	2.12	6.06	0.071
pH	**7.43**	**0.04**	(SD)	**7.44**	**0.05**	(SD)	**0.046 ***
pCO_2_ [mmHg]	32.75	29.15	35.05	30.40	27.00	33.50	0.069
pO_2_ [mmHg]	**65.80**	**59.20**	**70.95**	**71.20**	**65.00**	**76.40**	**0.006**
SatO_2_	**93.45**	**91.35**	**94.55**	**95.10**	**93.30**	**96.10**	**0.000**
24–59 mo	mean/median with SD or IQR			mean/median with SD or IQR			*p*
age [months]	37	29	48	39	31	50	0.878
duration of signs/syndromes [days]	**4**	**3**	**6**	**3**	**1**	**5**	**0.020**
the highest fever [Celsius degrees]	39.2	39.0	40.0	39.5	39.0	40.0	0.789
LOS [days]	**7**	**5**	**9**	**4**	**3**	**6**	**0.000**
breath rate [per minute]	**28**	**24**	**30**	**24**	**22**	**26**	**0.029**
heart rate [per minute]	120	100	130	109	100	120	0.137
CRP [mg/L]	7.70	3.46	16.64	5.75	1.00	17.21	0.086
PCT [ng/mL]	0.21	0.13	0.80	0.17	0.10	0.36	0.135
WBC [*10^3/μL]	8.09	5.81	9.95	7.52	5.42	10.47	0.270
ANC [*10^3/μL]	4.32	2.31	7.26	3.95	2.19	6.69	0.389
pH	7.43	0.04	(SD)	7.43	0.05	(SD)	0.974 *
pCO_2_ [mmHg]	32.60	4.94	(SD)	33.40	5.65	(SD)	0.847 *
pO_2_ [mmHg]	65.90	60.00	74.85	68.95	63.80	77.20	0.055
SatO_2_	93.70	91.20	95.40	94.25	92.60	95.90	0.054
>60 mo	mean/median with SD or IQR			mean/median with SD or IQR			*p*
age [months]	74	64	95	84	70	107	0.169
duration of signs/syndromes [days]	**5**	**2**	**6**	**3**	**1**	**4**	**0.010**
the highest fever [Celsius degrees]	39.5	39.0	40.0	39.5	39.0	40.0	0.958
LOS [days]	**6**	**4**	**8**	**4**	**3**	**5**	**0.001**
breath rate [per minute]	22	20	24	22	20	24	0.696
heart rate [per minute]	100	90	110	91	85	100	0.150
CRP [mg/L]	6.83	1.90	20.70	6.99	3.19	15.05	0.841
PCT [ng/mL]	0.22	0.10	0.45	0.14	0.08	0.26	0.104
WBC [*10^3/μL]	5.94	4.70	8.89	6.18	4.43	8.51	0.941
ANC [*10^3/μL]	3.56	2.09	6.89	3.52	2.28	5.42	0.961
pH	7.44	7.41	7.44	7.42	7.41	7.45	0.658
pCO_2_ [mmHg]	35.30	30.90	39.90	35.80	32.20	38.60	0.773
pO_2_ [mmHg]	67.60	59.70	74.30	69.70	63.70	75.00	0.288
SatO_2_	94.10	89.80	95.30	94.45	92.70	95.60	0.287

**Table 2 diagnostics-12-02412-t002:** The results of the ROC curve analysis for the prediction of a lower respiratory tract infection (LRTI) and radiologically confirmed pneumonia (CXR + pneumonia). The cut-off values were calculated with the Youden index. AUC—area under the curve, 95%CI—95% confidence interval, PPV—positive predictive value, NPV—negative predictive value.

**0–6 mo**									
**LRTI**	AUC	95%CI		*p*	cut off	Sensitivity 95%CI	Specificity 95%CI	PPV 95%CI	NPV 95%CI
pH	0.651	0.520	0.783	0.024	7.442	73.91%	54.17%	43.59%	81.25%
						51.59–89.77%	39.17–68.63%	34.31–53.35%	67.50–90.04%
pCO_2_	0.749	0.633	0.865	0.000	36.10	69.57%	72.92%	55.17%	83.33%
						47.08–86.79%	58.15–84.72%	41.84–67.80%	72.47–90.47%
pO_2_	0.727	0.604	0.849	0.000	58.00	73.91%	60.42%	47.22%	82.86%
						51.59% -89.77%	45.27–74.23%	36.89–57.79%	70.07–90.89%
SatO_2_	0.740	0.620	0.861	0.000	93.00	91.30%	50.00%	46.67%	92.31%
						71.96–98.93%	35.23–64.77%	39.10–54.39%	75.60–97.89%
**CXR+ pneumonia**	AUC	95%CI		*p*	cut off	Sensitivity 95%CI	Specificity 95%CI	PPV 95%CI	NPV 95%CI
pH	insignificant								
pCO_2_	0.740	0.607	0.873	0.000	36.10	73.33%	67.86%	37.93%	90.48%
						44.90–92.21%	54.04–79.71%	27.28–49.88%	80.11–95.73%
pO_2_	0.754	0.619	0.888	0.000	58.00	86.67%	58.93%	36.11%	94.29%
						59.54–98.34%	44.98–71.90%	28.05–45.03%	81.68–98.39%
SatO_2_	0.756	0.621	0.891	0.000	91.60	93.33%	51.79%	34.15%	96.67%
						68.05–99.83%	38.03–65.34%	27.69–41.25%	81.11–99.49%
**6–23 mo**									
**LRTI**	AUC	95%CI		*p*	cut off	Sensitivity 95%CI	Specificity 95%CI	PPV 95%CI	NPV 95%CI
pH	0.622	0.510	0.734	0.033	7.455	82.50%	47.37%	52.38%	79.41%
						67.22–92.66%	33.98–61.03%	45.28–59.39%	65.10–88.86%
pCO_2_	insignificant								
pO_2_	0.666	0.556	0.776	0.003	68.60	72.50%	59.65%	55.77%	75.56%
						56.11–85.40%	45.82–72.44%	46.58–64.58%	64.15–84.23%
SatO_2_	0.714	0.609	0.818	0.000	94.00	72.50%	64.91%	59.18%	77.08%
						56.11–85.40%	51.13–77.09%	49.25–68.42%	66.26–85.21%
**CXR+ pneumonia**	AUC	95%CI		*p*	cut off	Sensitivity 95%CI	Specificity 95%CI	PPV 95%CI	NPV 95%CI
pH	insignificant								
pCO_2_	0.640	0.527	0.754	0.016	28.00	92.86%	37.68%	37.68%	92.86%
						76.50–99.12%	26.29–50.17%	32.89–42.73%	76.77–98.08%
pO_2_	0.695	0.588	0.803	0.000	68.60	82.14%	57.97%	44.23%	88.89%
						63.11–93.94%	45.48–69.76%	36.39–52.37%	77.90–94.78%
SatO_2_	0.755	0.655	0.855	0.000	94.00	82.14%	62.32%	46.94%	89.58%
						63.11–93.94%	49.83–73.71%	38.42–55.64%	79.19–95.11%
**24–59 mo**									
**LRTI**	AUC	95%CI		*p*	cut off	Sensitivity 95%CI	Specificity 95%CI	PPV 95%CI	NPV 95%CI
pH	insignificant								
pCO_2_	insignificant								
pO_2_	0.601	0.501	0.702	0.048	76.60	91.30%	29.07%	40.78%	86.21%
						79.21–97.58%	19.78–39.86%	36.93–44.74%	69.84–94.40%
SatO_2_	0.602	0.502	0.702	0.045	96.20	95.65%	23.26%	40.00%	90.91%
						85.16–99.47%	14.82–33.61%	36.89–43.20%	70.97–97.61%
**CXR+ pneumonia**	AUC	95%CI		*p*	cut off	Sensitivity 95%CI	Specificity 95%CI	PPV 95%CI	NPV 95%CI
pH	insignificant								
pCO_2_									
pO_2_									
SatO_2_									

**Table 3 diagnostics-12-02412-t003:** The results of the ROC curve analysis for the performance of chest X-ray (CXR) and prolonged length of stay (LOS). The cut-off values are calculated with the Youden index. AUC—area under the curve, 95%CI—95% confidence interval, PPV—positive predictive value, NPV—negative predictive value.

**0–6 mo**											
**CXR**						**LOS**					
	AUC	95%CI		*p*	cut off		AUC	95%CI		*p*	cut off
pH	insignificant					pH	insignificant				
pCO_2_	0.663	0.519	0.808	0.027	36.1	pCO2	0.636	0.503	0.768	0.045	36.0
pO_2_	0.703	0.573	0.833	0.002	52.3	pO2	0.726	0.608	0.843	0.000	52.3
SatO_2_	0.676	0.533	0.819	0.016	91.6	SatO2	0.712	0.591	0.833	0.001	88.2
**6–23 mo**											
**CXR**						**LOS**					
	AUC	95%CI		*p*	cut off		AUC	95%CI		*p*	cut off
pH	insignificant					pH	insignificant				
pCO_2_	0.622	0.510	0.734	0.033	28.5	pCO2	insignificant				
pO_2_	0.693	0.587	0.799	0.000	68.6	pO2	0.633	0.517	0.750	0.025	68.1
SatO_2_	0.751	0.652	0.849	0.000	94.6	SatO2	0.659	0.544	0.775	0.007	92.9
**24–59 mo**											
**CXR**						**LOS**					
		AUC	95%CI	*p*	cut off		AUC	95%CI		*p*	cut off
pH	insignificant					pH	insignificant				
pCO_2_						pCO2	0.390	0.288	0.493	0.036	
pO_2_						pO2	insignificant				
SatO_2_						SatO2					

**Table 4 diagnostics-12-02412-t004:** The odds ratio of a negative CXR; a comparison between the theoretical CBG-driven model and a lack of CBG-driven protocol. The results are shown separately for the age groups and combined for the parameter (SpO_2_ or pO_2_) as odds ratio and the corresponding odds reduction. OR—odds ratio, 95%CI—95% confidence interval. Note: SatO_2_ and pO_2_ were selected as tools for the decision-making process due to the highest negative predictive value.

		Negative CXR				Odds Reduction of CXR Performance		
	Cut-Off	OR	95%CI		*p*	%	95%CI	
SatO_2_ < 6 mo	91.6%	0.1644	0.0781	0.3461	<0.01	83.56	65.39	92.19
SatO_2_ 6–23 mo	94%	0.1486	0.0793	0.2785	<0.01	85.14	72.15	92.07
SatO_2_ 0–23 mo		0.1585	0.0986	0.2550	<0.01	84.15	74.5	90.14
pO_2_ < 6 mo	58 mmHg	0.1283	0.0603	0.2734	<0.01	87.17	72.66	93.97
pO_2_ 6–23 mo	68.6 mmHg	0.1731	0.0933	0.3210	<0.01	82.69	67.9	90.67
pO_2_ 0–23 mo		0.1542	0.0958	0.2483	<0.01	84.58	75.17	90.42

## Data Availability

Data are available on request from the authors.

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
