# Peer review of "Capillary Blood Gas in Children Hospitalized Due to Influenza Predicts the Risk of Lower Respiratory Tract Infection"

_diagnostics, 2022, doi:10.3390/diagnostics12102412_

Round 1

Reviewer 1 Report

I've read an interesting article entitled Capillary blood gas in children hospitalized due to influenza predicts the risk of lower respiratory tract infection. Here are some comments and remarks for authors:

Abstract

Line 10-11 “Capillary blood gas (CBG) reflects arterial blood values but is a less invasive 10 method” – less invasive than what?

Table 1 – please insert a table not a figure on a table – it will be more readable. Also please explain more accurately what these numbers represents (if its mean or median or SD in rows 1,2 and 3)

Discussion: Please add some remarks regarding collection of CBG samples. It is easier to obtain reliable sample in a younger children and it should be also mentioned.

 In general, I suggest to limit the conclusion to the age group of less than 6 months, since the statistical power for this group is the strongest.

Author Response

Dear Reviewer!

We would like to express our gratitude for Your constructive comments and remarks that will improve this manuscript! We would like to address directly all the issues that You have mentioned (we left them in italics):

Line 10-11 “Capillary blood gas (CBG) reflects arterial blood values but is a less invasive 10 method” – less invasive than what?

- it is less invasive than arterial blood sampling, we corrected the sentence

Table 1 – please insert a table not a figure on a table – it will be more readable. Also please explain more accurately what these numbers represents (if its mean or median or SD in rows 1,2 and 3)

­-Certainly, originally, I put in a figure because of the template- the table is quite huge and it is easier to use figures instead of reformatting it. With regards to mean/median- in general the values presented in the table are median values with IQR, but since in few cases the data was distributed normally we presented mean and SD- it is marked with (SD) in corresponding rows, but in fact it is not so transparent. In order to facilitate the reading I marked means and SD with blue. I also put the table instead of figure.

Discussion: Please add some remarks regarding collection of CBG samples. It is easier to obtain reliable sample in a younger children and it should be also mentioned.

- Thank you for this comment! In fact, it is the most important message of this study- if there is a less invasive method that gives comparable results, it should be preferred. We added a paragraph in the beginning of the discussion section.

 In general, I suggest to limit the conclusion to the age group of less than 6 months, since the statistical power for this group is the strongest.

- I totally agree, that the highest power is seen in the group of children <6mo, however, due to practical implications (a huge number of patients aged <2 who are hospitalized every year) we would like to leave the conclusions as they are, especially since it is a single centre study and better results might be expected also in older children. We added a sentence: “children under 6 months of age are the group which would benefit the most from CBG”- this sentence underlines the statistical power of the results and practical implications, but at the same time does not exclude or underestimate the use of CBG in older children. We had a long discussion, mostly on older children- should we put the results into this analysis, and we agreed to do so- in case anyone would need to repeat or wish to check its usefulness also in older children.

Best regards,

The authors

Reviewer 2 Report

Overall, this is an interesting paper with good scientific merit. The potential clinical impact of this paper is clearly significant.

The authors performed very detailed statistical analyses and their discussion was solid.  

A good attempt was made to explain the reason the predictive value of the pCO2, pH, etc. gradually diminished with increasing age.

Important study limitations (e.g. single center setting, lack of influenza subtyping etc.) were mentioned.

References appear to be up to date, with many of them published within the past five years.

Author Response

Dear Reviewer,

Thank you very much for Your flattering opinion, we hope the study would focus interest on capillary blood gas use in children, since this is much less ivasive method of monitoring patients and is also possibly beneficial in other aspects, like a decresae in the number of unnecessary chest X-rays.

Best regards,

The authors